# Clinical characteristics and risk factors associated with severe community-acquired pneumonia infected by *Chlamydia psittaci*

Tingting Xu,[1] Qi Yuan,[1] Jiayue Wang,[1] Zhenzhen Wu,[1] Zhongqi Chen,[1] Zhengxia Wang,[1] Wenkui Sun,[1] Mingshun Zhang,[1] Ningfei Ji,[1] Mao Huang[1]

**ABSTRACT**   The study aimed to describe the clinical features of psittacosis pneumonia and identify risk factors associated with severe psittacosis pneumonia. We retrospectively collected data from patients with psittacosis pneumonia, confirmed by metagenomic next-generation sequencing (mNGS) from January 2022 to June 2024 and analyzed differences between severe and non-severe groups. A total of 24 patients (eight severe/16 non-severe) were diagnosed with psittacosis, and 33.3% had severe psittacosis pneumonia. The levels of white blood cells (WBCs), neutrophils, neutrophil-to-lymphocyte ratio (NLR), hypersensitive C-reactive protein (hs-CRP), and procalcitonin (PCT) and the detection rate of fungi by mNGS were significantly higher in the severe group than in the non-severe group. The length of stay and omadacycline use were longer in patients of the severe group when compared to the non-severe group. Receiver operating characteristic (ROC) curves indicated that NLR had a better predictive value of severe conditions than PCT and hs-CRP. Patients with psittacosis pneumonia have a high rate of severe disease, and NLR may be a useful tool to promptly ascertain the severity of the disease and optimal therapies for better outcomes.

**IMPORTANCE** This study explores the clinical features of psittacosis pneumonia and identifies key factors that may predict the severity of the disease. By analyzing data from 24 patients diagnosed using metagenomic next-generation sequencing (mNGS), the research uncovers important differences between severe and non-severe cases. The study finds that patients with severe psittacosis pneumonia have elevated levels of white blood cells (WBCs), neutrophils, neutrophil-to-lymphocyte ratio (NLR), C-reactive protein (hs-CRP), and procalcitonin (PCT), as well as a higher detection rate of fungi. Notably, the NLR emerges as a strong predictor of severe disease, suggesting its potential as an early diagnostic tool. These findings provide valuable insights that can help healthcare providers identify high-risk patients more quickly, allowing for timely interventions and improved management of the disease. Ultimately, this research could lead to better outcomes by guiding treatment decisions and enhancing our understanding of psittacosis pneumonia.

**KEYWORDS**   *Chlamydia psittaci*, severe pneumonia, neutrophil-to-lymphocyte ratio, metagenomic next-generation sequencing, procalcitonin

Psittacosis pneumonia is an infectious disease caused by *Chlamydia psittaci*, an intracellular bacterium that often infects birds and ducks and has subsequently been transmitted to human beings (1). The prevalence of chlamydial infection in birds was 19.5%, while the infection rate of psittacosis pneumonia varied a lot due to different climates, temperatures, jobs, exposure, and dietary habits (2, 3). *Chlamydia psittaci* is a gram-negative bacterium with biphase development, including reticular body (RB)-to-elementary body (EB) differentiation, and the pathologic lesions for human beings

Address correspondence to Ningfei Ji, jiningfei@163.com, or Mao Huang, hm6114@163.com.

Tingting Xu and Qi Yuan contributed equally to this article. The author order was determined based on their contributions to the study.

The authors declare no conflict of interest.

See the funding table on p. 14.

are primarily determined by the host's immune response to *Chlamydia psittaci* (1, 4). Patients are infected with *Chlamydia psittaci* via inhalation of contaminated aerosols. The pathogen initially adheres to respiratory epithelial cells, subsequently establishes an intracellular niche within mononuclear phagocytes, and exploits this compartment to evade host immune defenses and lysosomal degradation (5). The clinical signs of psittacosis among individuals varied from asymptomatic infection to severe pneumonia or even death because of widespread epithelial and macrophage infection in many organs (6, 7). The nonspecific nature of symptoms overlaps with those of influenza, Corona Virus Disease 2019 (COVID-19), and other respiratory pathogens, leading to a high clinical misdiagnosis rate of psittacosis of 50%–80% (8, 9).

The traditional diagnostic methods of psittacosis, such as the culture of *Chlamydia psittaci* and polymerase chain reaction (PCR), are limited. Therefore, a highly sensitive and broad-spectrum approach for detecting *Chlamydia psittaci* nucleic acids, termed metagenomic next-generation sequencing (mNGS), has been increasingly applied in the diagnosis of infectious diseases over the past decades (7, 10). The appropriate drugs for *Chlamydia psittaci* include tetracyclines, macrolides, and quinolones (11).

Recently, many studies have focused on the clinical characteristics of psittacosis in different countries and regions, without a consistent conclusion among these cohorts (11–13). Duck is a natural host for *Chlamydia psittaci*. As reported, people in Nanjing could consume 100 million ducks a year, ranking among the highest in China. Hence, we conducted a retrospective study to analyze the clinical features of psittacosis and compare patients with severe psittacosis pneumonia to those with non-severe conditions who were diagnosed by mNGS in the First Affiliated Hospital of Nanjing Medical University. This study aimed to investigate the clinical characteristics and risk factors for the severity of psittacosis.

## MATERIALS AND METHODS

### Study design and participants

In this retrospective, single-center study, all enrolled patients were diagnosed with psittacosis pneumonia and admitted to the First Affiliated Hospital of Nanjing Medical University from January 2022 to June 2024. Written informed consent was waived because this study was retrospective in nature. The study was conducted following the guidelines of Helsinki Declaration. All participants were diagnosed and grouped according to the American Thoracic Society (ATS)/Infectious Diseases Society of America (IDSA) guidelines

### The workflow of mNGS (sample processing, DNA/RNA extraction, construction of DNA/RNA libraries, sequencing, and bioinformatics analysis)

The mNGS was performed as previously described (10, 14–16). Briefly, BALF samples (2–3 mL) were collected from the corresponding sites of lung lesions of patients via a bronchoscope (OLYMPUS). BALF samples were instantaneously centrifuged, and mechanical disruption with beads was performed (Hangzhou Matrix Biotechnology Co., Ltd.). Genomic DNA and RNA were extracted from the specimens using the NGSmaster automatic nucleic acid detection reaction and construction system. Sequencing libraries of a mixture of DNA and RNA pathogens were constructed through reverse transcription of the RNA to complementary DNA (cDNA) by using the SuperScript Double-Stranded cDNA Synthesis Kit (11917020, Invitrogen), followed by nucleic acid fragmentation, end-repair, terminal adenylation, adapter ligation, and purification. The quality of the libraries was assessed using the Qubit 3.0 platform (Thermo Fisher Scientific, USA) and the Agilent 2100 Bioanalyzer Instrument (Agilent Technologies). The sequencing of libraries was performed on an Illumina NextSeq 550.

Bioinformatics analysis of pathogenic microbial gene data was automatically processed on the Gentellix software to produce a detection report, including the

elimination of low-quality or undetected sequences, high-coverage repeats, splice contaminants, and short-read-length sequences. To remove human host sequences, the sequencing data were compared to the human reference genome (GRCh38.p13). The remaining sequences were aligned to NCBI GenBank and a previously constructed reference database to identify the pathogens; the identified reads and their relative abundance were normalized to the number of reads per ten million (RPTM) to determine positive results. The relative abundance refers to the distribution proportion of a microbial sequence in the five major microorganisms after removal of the host sequence in each sample. The criteria for positive mNGS were as follows: (i) the total sequencing number of each specimen was at least 20 million reads. (ii) For common species such as viruses, bacteria (excluding mycobacteria), and parasites, the coverage rate of these identified microbes was ten-fold greater than that of other microbes. (iii) The coverage rate of fungi, such as *Aspergillus* and *Candida*, was fivefold higher than that of other fungi due to their low biomass and the difficulty encountered during DNA extraction.

## Criteria for diagnosis and grouping

The inclusion criteria for psittacosis pneumonia are as follows: (i) patients meeting the diagnostic criteria for community-acquired pneumonia (CAP) according to the ATS/IDSA guidelines. (ii) specific *Chlamydia psittaci* DNA fragments detected by mNGS in serial samples of bronchi-alveolar lavage fluid (BALF), sputum, or blood. Regarding sample selection, bronchoalveolar lavage fluid (BALF) served as the primary specimen type. However, two patients declined BALF collection and provided only blood samples. One patient contributed BALF, blood, and sputum specimens, while another provided paired BALF and blood samples. The remaining 20 patients were diagnostically evaluated exclusively through BALF analysis. Patients with incomplete data were excluded.

Severe pneumonia was defined as a patient who met either one major criterion or three or more minor criteria based on the ATS/IDSA CAP guidelines (17). The major criteria were (i) respiratory failure requiring mechanical ventilation and (ii) septic shock with the need for vasopressors. Minor criteria were (i) respiratory rate 30 breaths/min; (ii) PaO2/FiO2 250 mmHg; (iii) multilobar infiltrations; (iv) confusion/disorientation; (v) blood urea nitrogen level 7.14 mmol/L; (vi) white blood cell count 4000 cell/; (vii) platelet count 100,000/; (viii) core temperature 36°C; (ix) systolic blood pressure 90 mmHg requiring aggressive fluid resuscitation.

## Data collection

Data on clinical symptoms, dynamics, previous history, laboratory tests, chest computed tomography (CT), and mNGS reports were extracted from electronic medical records. The diagnosis of pathogens was identified by clinicians based on clinical features, laboratory examinations, and imaging. In addition, mNGS was performed as described previously (16). In short, samples including bronchoalveolar lavage fluid (BALF), sputum, and blood were collected, the nucleic acid was extracted, the deoxyribonucleic acid (DNA) libraries were constructed, and sequencing was performed and analyzed by bioinformatics methods. Additional information on treatments, outcomes, and relevant follow-up imaging was also collected.

## Statistical analysis

All data were analyzed using SPSS version 26.0 software and GraphPad Prism 8.0 software. For continuous variables with normal distribution, the data were expressed as mean ± standard deviation, and the independent sample *t*-test was applied between the two groups. For continuous variables with nonnormal distribution, the data were described as median (interquartile range [IQR]) and analyzed with the Mann-Whitney U test. For categorical variables, the data were expressed as numbers (percentages) and analyzed by Pearson's $\chi^2$ or Fisher's exact tests. Missing values exhibited nonrandom patterns aligned with clinical protocols (e.g., stable patients omitting blood gas

analysis; non-diabetics foregoing HbA1c testing). Consequently, these variables were retained in univariate analyses but excluded through listwise deletion in multivariate regression. For outliers (>3 × IQR beyond quartiles or exceeding clinical thresholds), all instances underwent blinded verification by two clinicians. Biologically plausible outliers were retained and analyzed via Huber regression to minimize leverage. Comprehensive robustness analyses confirmed consistent conclusions across all methodological approaches. Pearson's correlation analysis was conducted to evaluate the correlations among neutrophil-to-lymphocyte ratio (NLR), procalcitonin (PCT), C-reactive protein (hs-CRP), alanine aminotransferase (ALT), aspartate transaminase (AST), and the number of pathogens. The reads and relative abundance of *Chlamydia psittaci* exhibited non-normal distributions. Clinical outcomes (severe pneumonia status and post-discharge transfer to secondary hospitals) were dichotomized. Spearman's rank correlation analysis was performed to analyze the associations between *Chlamydia psittaci* parameters (reads, relative abundance) and inflammatory markers (NLR, PCT, and hs-CRP) or clinical outcomes, with correlation coefficients reported as Spearman's Rho (ρ). The receiver operating characteristic (ROC) curves and area under the curves (AUC) were used to compare the value of inflammatory markers in predicting the severity of psittacosis pneumonia. A two-tailed *P*-value < 0.05 was considered statistically significant.

## RESULTS

### General characteristics of the enrolled patients

A total of 24 patients were finally included in this study, and the incidence of severe pneumonia was about 33.3%. The basic characteristics and clinical symptoms of the enrolled patients are listed in Table 1. Approximately half of the patients were mainly infected in winter (12 patients, 50.0%) with exposure to duck, dove, chicken, and parrot (45.80%). A significant difference was observed in the length of stay between severe and non-severe groups, whereas there were no significant differences in age, seasons, exposure, and incubation period between the severe and non-severe groups.

All patients had a fever (temperature ranging from 38°C to 41°C). Other common symptoms were cough (79.20%), dyspnea (58.30%), fatigue (60.90%), and nervous signs, including headache (50.00%) and consciousness (24.20%). However, there was no significant difference in clinical signs between the severe and non-severe groups, in terms of symptoms related to respiratory (cough was 87.50% vs 75.00%; the expectation was 50.00% vs 37.50%; dyspnea was 62.50% vs 56.30%), nervous system (headache was 50.00% vs 50.00%; consciousness was 37.50% vs 25.00%), and gastric system (vomiting was 50.00% vs 25.00%; diarrhea was 0.00% vs 12.50%).

### Comparison of laboratory examinations between the two groups

The laboratory test results of the enrolled patients are detailed in Table 2. Patients with pneumonia in the severe group exhibited high levels of white blood cells (WBCs), neutrophil, NLR, hs-CRP, and PCT compared to the non-severe group (*P* < 0.05). Notably, as shown in Fig. 1, Hb and PaO2/FiO2 were dramatically lower in the severe group than in the non-severe group. No significant difference was observed in the levels of lymphocytes, platelets, pH, D-Dimer, estimated glomerular filtration rate (eGFR), fibrinogen (FIB), ALT, AST, glucose, free triiodothyronine (FT3), free tetraiodothyronine (FT4), and thyroid-stimulating hormone (TSH).

### Radiographic findings

All subjects underwent CT scans during admission. The radiographic features for the two groups are listed in Table 3. Patients developed mainly lobar pneumonia. The lesions were bilateral in 10 cases (41.70%) and unilateral in 14 cases (58.30%), including nine cases on the left and five cases on the right. There was no significant difference in the distribution of lesions between the severe and non-severe groups. The pleural

**TABLE 1** Basic information and clinical symptoms of enrolled patients[a]

| Variables | Total | Severe | Non-severe | Statistics | P value |
|---|---|---|---|---|---|
| | (n = 24) | (n = 8) | (n = 16) | | |
| Male, n (%) | 14 (58.30%) | 4 (50.00%) | 10 (62.50%) | 0.343 | 0.673 |
| Age mean ± SD | 55.71 ± 13.43 | 62.50 ± 7.65 | 52.31 ± 14.57 | 1.840 | 0.079 |
| Season | | | | 2.100 | 0.552 |
| Spring, n (%) | 3 (12.80%) | 1 (12.50%) | 2 (12.50%) | | |
| Summer, n (%) | 4 (16.70%) | 1 (12.50%) | 3 (18.75%) | | |
| Autumn, n (%) | 5 (20.80%) | 3 (37.50%) | 2 (12.50%) | | |
| Winter, n (%) | 12 (50.00%) | 3 (37.50%) | 9 (56.25%) | | |
| History of exposure, n (%) | 11 (45.80%) | 3 (37.50%) | 8 (50.00%) | 0.336 | 0.679 |
| Duck | 4 (16.70%) | 2 (25.00%) | 2 (12.50%) | | |
| Dove | 2 (8.30%) | 1 (12.50%) | 1 (6.25%) | | |
| Chicken | 1 (4.20%) | 0 (0.00) | 1 (6.25%) | | |
| Parrot | 4 (16.70%) | 0 (0.00) | 4 (25.00%) | | |
| The days from clinical symptoms to admission median (IQR) | 9.75 (8.75) | 12.00 (10.25) | 8.50 (4.5) | 51.500 | 0.458 |
| Length of stay (days) | 12.88 ± 6.52 | 18.00 ± 6.07 | 10.31 ± 5.20 | 3.233 | 0.004 |
| Fever | 24 (100%) | 8 (100%) | 16 (100%) | 0.000 | 1.000 |
| Cough | 19 (79.20%) | 7 (87.50%) | 12 (75.00%) | 0.505 | 0.631 |
| Expectorations | 10 (41.70%) | 4 (50.00%) | 6 (37.50%) | 0.343 | 0.673 |
| Dyspnea | 14 (58.30%) | 5 (62.50%) | 9(56.30%) | 0.086 | 1.000 |
| Chest pain | 2 (8.30%) | 2 (25.00%) | 0 (0.00%) | 4.364 | 0.101 |
| Fatigue | 14 (60.90%) | 3 (37.50%) | 11 (73.30%) | 2.813 | 0.179 |
| Headache | 12 (50.00%) | 8 (50.00%) | 4 (50.00%) | 0.000 | 1.000 |
| Consciousness | 7 (24.20%) | 3 (37.50%) | 4 (25.00%) | 0.403 | 0.647 |
| Vomit | 8 (33.30%) | 4 (50.00%) | 4 (25.00%) | 1.500 | 0.363 |
| Diarrhea | 2 (8.30%) | 0 (0.00%) | 2 (12.50%) | 1.091 | 0.536 |

[a]SD, standard deviation; IQR, interquartile range.

effusion occurred in 20 cases (83.30%), and 14 cases (58.30%) were bilateral effusions. No significant difference was observed in the pleural effusion between the severe group and the non-severe group.

## Co-infection with other pathogens in the severe and non-severe pneumonia groups

A total of 24 patients were identified with *Chlamydia psittaci* infection using mNGS; the reads and abundance varied a lot. As exhibited in Table 4, the average number of pathogens detected by mNGS in the severe group (4.38 ± 2.32) was significantly more than that of the non-severe group (1.94 ± 1.39). Among 24 patients, 15 cases (62.50%) were suspected to be co-infected with other pathogens; however, there was no significant difference in the detection of coinfections or the detection rates of bacteria and viruses. Notably, mNGS results indicated that the detection rate of fungi in the severe group (50.00%) was significantly higher than that in the non-severe group (6.30%). The incidence of fungi co-infection in the severe group was mainly related to *Candida albicans* (3/37.50%, *Aspergillus* (3/37.50%), and *Pneumocystis jirovecii* (1/12.50%). The proportion of co-infected microorganisms in the detection of mNGS is illustrated in Fig. 2, and the details of antifungal treatments for these patients are listed in Table 5.

## Treatment and prognosis

In terms of comorbidities, enrolled patients tended to exhibit one or more dysfunctions. As shown in Table 4, higher hyperfibrinogenemia, liver injury, and hypoalbuminemia were the top three disorders among all patients, which were higher than 50.00%. No significant difference was observed in the occurrence of hypertension, diabetes, cerebral

**TABLE 2** Laboratory examination results[a]

| Variables | Total (n = 24) | Missing | Severe (n = 8) | Non-severe (n = 16) | Statistics | P value |
|---|---|---|---|---|---|---|
| WBC | 9.00 ± 5.45 | 0 | 12.75 ± 3.73 | 7.12 ± 5.26 | 2.693 | 0.013 |
| Neutrophil | 7.68 ± 5.43 | 0 | 11.42 ± 3.54 | 5.8 ± 5.30 | 2.697 | 0.013 |
| Lymphocyte | 0.80 ± 0.43 | 0 | 0.66 ± 0.20 | 0.87 ± 0.50 | 1.122 | 0.274 |
| NLR | 12.83 ± 10.17 | 0 | 18.99 ± 9.21 | 9.76 ± 9.42 | 2.280 | 0.033 |
| Hemoglobin | 112.67 ± 19.75 | 0 | 99.75 ± 20.29 | 119.13 ± 16.52 | 2.513 | 0.020 |
| Platelet | 220.63 ± 109.53 | 0 | 193.75 ± 114.80 | 234.06 ± 108.01 | 0.821 | 0.407 |
| pH | 7.52 ± 0.08 | 7 | 7.52 ± 0.11 | 7.50 ± 0.05 | 0.558 | 0.585 |
| PaO2 | 66.25 ± 17.34 | 9 | 57.96 ± 18.95 | 73.50 ± 12.83 | 0.595 | 0.082 |
| PaO2/FiO2 | 241.18 ± 110.36 | 9 | 178.65 ± 85.76 | 295.89 ± 103.56 | 2.366 | 0.034 |
| PaCO2 | 31.38 ± 8.41 | 7 | 34.13 ± 10.20 | 28.94 ± 6.01 | 1.294 | 0.215 |
| D-Dimer | 5.65 ± 8.53 | 3 | 4.38 ± 2.64 | 6.44 ± 10.74 | 0.529 | 0.603 |
| FIB | 6.43 ± 2.23 | 0 | 6.29 ± 2.21 | 6.49 ± 2.30 | 0.196 | 0.846 |
| hs-CRP | 157.37 ± 102.69 | 0 | 230.30 ± 101.75 | 120.90 ± 84.17 | 2.800 | 0.010 |
| PCT | 1.51 ± 2.01 | 1 | 2.79 ± 2.83 | 0.83 ± 0.98 | 2.463 | 0.023 |
| ALT | 108.53 ± 54.31 | 0 | 120.85 ± 56.83 | 102.36 ± 53.81 | 0.779 | 0.444 |
| AST | 123.54 ± 87.85 | 0 | 146.06 ± 97.99 | 112.28 ± 83.36 | 0.884 | 0.386 |
| Glucose | 7.61 ± 2.65 | 0 | 8.68 ± 3.17 | 7.06 ± 2.27 | 1.437 | 0.165 |
| HbA1c | 6.28% ± 0.72% | 9 | 6.27% ± 0.75% | 6.28% ± 0.74% | 0.009 | 0.993 |
| eGFR | 90.44 ± 29.88 | 0 | 75.20 ± 25.91 | 98.06 ± 29.50 | 1.858 | 0.077 |
| FT3 | 2.45 ± 1.23 | 8 | 2.24 ± 0.85 | 2.57 ± 1.44 | 0.513 | 0.616 |
| FT4 | 19.87 ± 19.77 | 8 | 15.72 ± 3.87 | 22.36 ± 25.00 | 0.637 | 0.534 |
| TSH | 1.97 ± 3.41 | 8 | 1.25 ± 1.54 | 2.40 ± 4.18 | 0.640 | 0.533 |

[a]WBC, white blood cells; NLR, neutrophil-to-lymphocyte ratio; FIB, fibrinogen; hsCRP, hypersensitive C-reactive protein; PCT, procalcitonin; ALT, alanine aminotransferase; AST, aspartate transaminase; HbA1c, Hemoglobin A1C; eGFR, estimated glomerular filtration rate; FT3, free triiodothyronine; FT4, free tetraiodothyronine; TSH, thyroid-stimulating hormone.

infarction, kidney injury, anemia, hyperfibrinogenemia, and hypoalbuminemia, while cardiac insufficiency was significantly higher in the severe group than in the non-severe group.

Oxygen and other life support were important strategies in maintaining the whole body. About 70.00% of patients received at least one kind of oxygen therapy. In the severe group, 87.50% received invasive ventilation. In the non-severe group, 50.00% inhaled oxygen by nasal catheter. Notably, two severe cases were supported by venovenous extracorporeal membrane oxygenation (V-V ECMO) for maintaining their respiratory and circulatory balance.

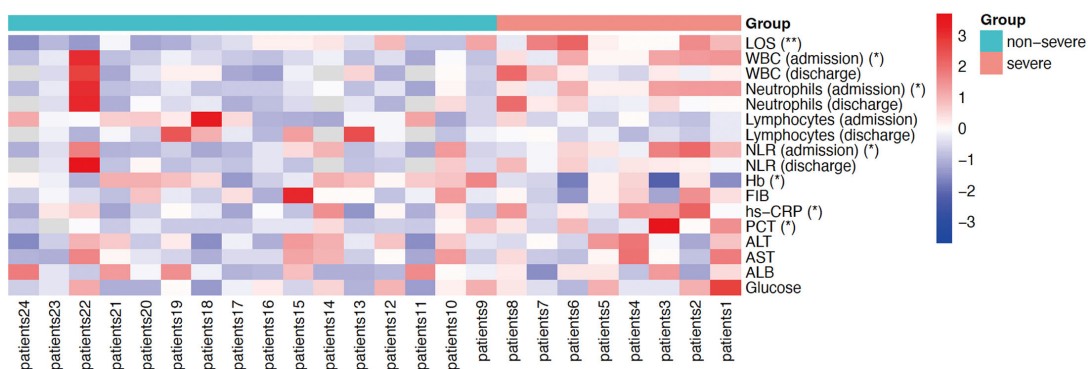

**FIG 1** Heatmap showing clinical blood examinations of 27 patients. The X-axis represents the patient, with red representing the severe group and green representing the non-severe group. The Y-axis was clinical blood tests, with redder patches symbolizing higher expression and bluer patches symbolizing lower expression. Statistical significance was analyzed between the severe and non-severe groups, denoted by *P < 0.05.

**TABLE 3** Radiographic features

| Variables | Total (n = 24) | Severe (n = 8) | Non-severe (n = 16) | Statistics | P value |
|---|---|---|---|---|---|
| Distribution of lesions | | | | 3.200 | 0.202 |
| Bilateral involvement | 10 (41.70%) | 5 (62.50%) | 5 (31.30%) | | |
| Unilateral, left lung | 9 (37.50%) | 3 (37.50%) | 6 (37.50%) | | |
| Unilateral, right lung | 5 (20.80%) | 0 (0.00%) | 5 (31.30%) | | |
| Pleural effusion | 20 (83.30%) | 8 (100.00%) | 12 (75.00%) | 4.500 | 0.105 |
| Unilateral effusion | 6 (25.00%) | 1 (12.50%) | 5 (31.30%) | | |
| Bilateral effusion | 14 (58.30%) | 7 (87.50%) | 7 (43.80%) | | |

Antibiotics are essential for treating psittacosis. The median days of antibiotics were 24 days and were comparable between the two groups. Over half of the patients were prescribed minocycline, and half of the patients were prescribed omadacycline. About a quarter of patients were prescribed moxifloxacin, and a fifth of cases were prescribed azithromycin. The frequency of drug administration was comparable between the two groups. However, the mean days of omadacycline in the severe group (11 days) was significantly longer than that in the non-severe group (4 days).

Among 24 patients, all patients recovered well, only a fifth of patients were transferred to primary hospitals, and the rest were discharged with oral prescriptions or without any drugs. Compared with the non-severe group (6.30%), the proportion of patients transferring to primary hospitals in the severe group was higher (50.00%).

## Predictive factors for severe pneumonia with *Chlamydia psittaci*

To identify representative markers to judge the status of patients with psittacosis, Pearson's correlation analysis was performed to evaluate the association among common laboratory examinations. Common inflammatory markers that were significantly different in this study included WBC, neutrophils, NLR, hs-CRP, and PCT (all $P < 0.05$). Concerning the WBC and neutrophils being closely related to NLR, we employed NLR to exhibit the inflammation status of peripheral blood. The correlation coefficients are displayed in Fig. 3. The correlation analysis showed that NLR was positively related to hs-CRP ($r = 0.68$, $P < 0.05$), PCT ($r = 0.60$, $P = 0.003$), ALT ($r = 0.68$, $P < 0.05$), and AST ($r = 0.51$, $P = 0.011$), while PCT was positively related with the numbers of pathogens (No.p) detected by mNGS ($r = 0.63$, $P = 0.001$), indicating the NLR and PCT may be a valuable marker to reflect the status of disease. In addition, Spearman's correlation analysis was conducted to explore the association between *Chlamydia psittaci* sequencing metrics (reads and relative abundance) and clinical variables. The correlation coefficients ($\rho$) are exhibited in Table 6. The correlation analysis revealed that both reads and relative abundance of *Chlamydia psittaci* were positively correlated with NLR (reads: $\rho = 0.46$, $P = 0.032$; relative abundance: $\rho = 0.44$, $P = 0.040$). Notably, the relative abundance also showed a positive correlation with the clinical endpoint of post-discharge transfer to secondary hospitals ($\rho = 0.45$, $P = 0.034$).

Thus, we employed the ROC curve to evaluate and compare the predictive values of different inflammatory markers for the severity of patients who were infected with *Chlamydia psittaci*. As shown in Fig. 4, the NLR showed a better effect in predicting severe pneumonia, with a maximum AUC (area under the curve) of 0.8125 ($P < 0.05$). PCT seems better than hs-CRP for predicting the severity of disease, with an AUC of 0.8083 ($P < 0.05$) and 0.7734 ($P < 0.05$), respectively.

Concerning the NLR may be a confounding factor to the ATS/IDSA CAP criteria, a logistic regression model was applied to explore independent factors associated with the severe group. Since the sample size of our study was relatively limited, the variables involved in the regression model should be fewer than three. The goodness of fit was judged by Nagelkerke $R^2$ and the Hosmer-Lemeshow test (H-L test). As summarized in Table 7, the results revealed that NLR was an independent risk factor for predicting the severe group (OR = 1.103, 95% CI 1.001–1.216, $P < 0.05$)

**TABLE 4** Diagnosis, treatment, and prognosis[a]

| Variables | Total (n = 24) | Severe (n = 8) | Non-severe (n = 16) | Statistics | P value |
|---|---|---|---|---|---|
| Sample BALF | 22 (91.70%) | 8 (100%) | 14 (87.50%) | 1.091 | 0.536 |
| Sample blood | 4 (16.70%) | 2 (25.00%) | 2 (12.50%) | 0.600 | 0.578 |
| Sample sputum | 1 (4.20%) | 1 (12.50%) | 0 (0.00%) | 2.087 | 0.333 |
| Days from the onset of symptoms to mNGS confirmation | 10 (6.0) | 8 (8.50) | 11 (5.00) | 44.500 | 0.238 |
| Number of pathogens reported by mNGS | 2.75 ± 2.07 | 4.38 ± 2.32 | 1.94 ± 1.39 | 3.230 | 0.004 |
| Mixed infection | 15 (62.50%) | 7 (87.50%) | 8 (50.00%) | 3.200 | 0.178 |
| Accompanied by bacteria | 8 (33.30%) | 4 (50.00%) | 4 (25.00%) | 1.500 | 0.221 |
| Accompanied by viruses | 10 (41.70%) | 5 (62.50%) | 5 (31.30%) | 2.143 | 0.204 |
| Accompanied by fungi | 5 (20.80%) | 4 (50.00%) | 1 (6.30%) | 6.189 | 0.028 |
| Comorbidity | | | | | |
| Cardiac insufficiency | 9 (37.50%) | 6 (75.00%) | 3 (18.80%) | 7.200 | 0.021 |
| Diabetes | 5 (20.80%) | 3 (37.50%) | 2 (12.50%) | | |
| Cerebral infarction | 5 (20.80%) | 3 (37.50%) | 2 (12.50%) | 2.021 | 0.289 |
| Hypertension | 9 (37.50%) | 4 (50.00%) | 5 (31.30%) | 0.800 | 0.412 |
| Liver damage | 20 (83.30%) | 8 (100%) | 12 (75.00%) | 2.400 | 0.262 |
| Kidney damage | 5 (20.80%) | 2 (25.00%) | 3 (18.80%) | 0.126 | 1.000 |
| Anemia | 11 (45.80%) | 5 (62.50%) | 6 (37.50%) | 1.343 | 0.390 |
| Hyperfibrinogenemia | 22 (91.70%) | 6 (75.00%) | 16 (100%) | 4.364 | 0.101 |
| Hypoalbuminemia | 17 (70.80%) | 6 (75.00%) | 11 (68.80%) | 0.101 | 1.000 |
| Treatment | | | | | |
| Oxygen interventions, n (%) | 17 (70.80%) | 8 (100%) | 9 (56.30%) | 4.941 | 0.054 |
| V-V ECMO | 2 (8.30%) | 2 (25.00%) | 0 (0.00) | 4.364 | 0.101 |
| Invasive mechanical ventilation | 7 (29.20%) | 7 (87.50%) | 0 (0.00) | 19.765 | 0.000 |
| Nasal catheter | 9 (37.50%) | 1 (12.50%) | 8 (50.00%) | 3.200 | 0.178 |
| HFNC | 2 (8.33%) | 0 (0.00) | 2 (12.50%) | 1.091 | 0.536 |
| Venturi mask | 1 (4.20%) | 0 (0.00) | 1 (6.30%) | 0.522 | 1.000 |
| Total days of antibiotics | 24.0 (10.75) | 24.5 (27.75) | 24.0 (9.75) | 56.500 | 0.6631 |
| Azithromycin use n (%) | 5 (20.80%) | 1 (12.50%) | 4 (25.00%) | 0.505 | 0.631 |
| Days of azithromycin | 2.79 ± 6.23 | 2.25 ± 6.36 | 3.06 ± 6.35 | 0.295 | 0.771 |
| Minocycline use n (%) | 14 (58.30%) | 5 (62.50%) | 9 (56.30%) | 0.086 | 1.000 |
| Days of minocycline | 3.46 ± 6.00 | 5.25 ± 8.38 | 2.56 ± 4.46 | 1.036 | 0.311 |
| Omadacycline use n (%) | 12 (50.00%) | 6 (75.00%) | 6 (37.50%) | 3.000 | 0.193 |
| Days of omadacycline | 6.08 ± 6.95 | 11.13 ± 7.59 | 3.56 ± 5.19 | 2.884 | 0.009 |
| Doxycycline use n (%) | 2 (8.30%) | 1 (12.50%) | 1 (6.30%) | 0.273 | 1.000 |
| Days of doxycycline | 0 (0.00) | 0 (0.00) | 0 (0.00) | 68.500 | 0.787 |
| Levofloxacin use n (%) | 2 (8.30%) | 1 (12.50%) | 1 (6.30%) | 0.273 | 1.000 |
| Days of levofloxacin | 0 (0.00) | 0 (0.00) | 0 (0.00) | 68.500 | 0.787 |
| Moxifloxacin use n (%) | 6 (25.00%) | 0 (0.00) | 6 (37.50%) | 4.000 | 0.066 |
| Days of moxifloxacin | 0 (3.00) | 0 (0.00) | 0 (5.75) | 40.000 | 0.153 |
| Outcomes | | | | 6.225 | 0.044 |
| Hospital survival | 24 (100.00%) | 8 (100.00%) | 16 (100.00%) | | |
| Discharge with oral drugs | 15 (62.50%) | 3 (37.50%) | 12 (75.00%) | | |
| Transfer to secondary hospitals | 5 (20.80%) | 4 (50.00%) | 1 (6.30%) | | |
| Discharge without interventions | 4 (16.70%) | 1 (12.50%) | 3 (18.80%) | | |

[a]mNGS, metagenomic next-generation sequencing; BALF, bronchi-alveolar lavage fluid; V-V ECMO, veno-venous extracorporeal membrane oxygenation; HFNC, high-flow nasal cannula.

## Follow-up of radiology

In terms of follow-up, 12 patients underwent CT scans 15–180 days after discharge. The primary lesions of most patients were nearly completely absorbed, and the primary lesions and the pleural infiltration of the rest were absorbed to some degree. However, patients with severe pneumonia tended to exhibit residual cord shadow and fibrosis

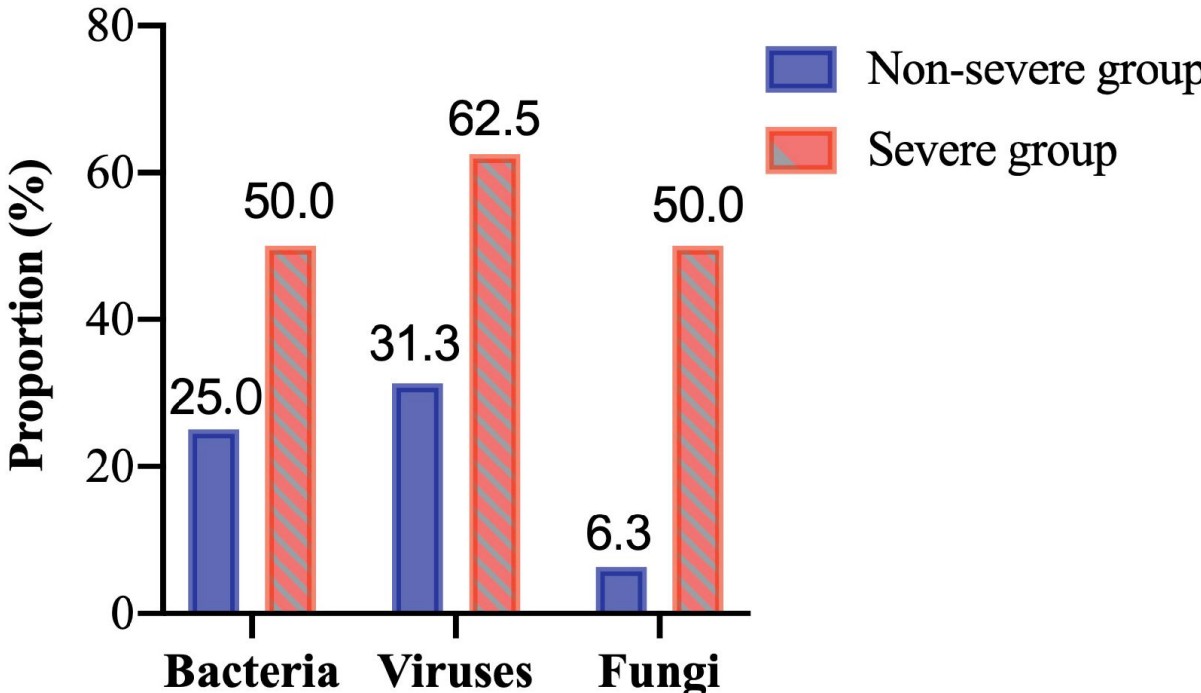

**FIG 2** The proportion of co-infected microorganisms in the detection of mNGS. The X-axis represents the type of co-infected microorganisms, including bacteria, viruses, and fungi, between the severe and non-severe groups, with the blue column symbolizing the non-severe group and the red column symbolizing the severe group. The Y-axis was the proportion of co-infection with a specific kind of microorganism in the detection of mNGS.

without extra treatment. Some representative pictures before and after treatment are shown in Fig. 5.

## DISCUSSION

In this study, we found that patients with psittacosis tended to be infected in autumn and winter after exposure to parrots and ducks. Besides, patients exhibited a series of symptoms including fever, cough, vomiting, and altered consciousness. Laboratory tests revealed that psittacosis affected the inflammation level, hematopoietic system, coagulation system, endocrine system, liver function, and kidney function, accompanied by multiple pathogens. Radiology showed various lesions in the lungs and pleural infiltration. In terms of treatment, tetracycline is a common strategy, and patients recover well after appropriate treatment. Compared with the non-severe group, patients with psittacosis in the severe group exhibited longer time of hospitalization; higher levels of WBC, hs-CRP, PCT, and NLR; higher proportions of co-infection with fungi; and longer prescriptions of omadacycline. Additionally, NLR showed a better value for predicting the severity of disease than CPR and PCT.

**TABLE 5** Anti-fungal treatments in cases of co-infection with fungi

| Case | Group | Fungi | Drugs and days of antifungal treatment |
|---|---|---|---|
| 1 | Severe | *Aspergillus flavus, Aspergillus fumigatus*, and *Candida albicans* | Voriconazole injection for 5 days |
| 2 | Severe | *Aspergillus flavus, Aspergillus oryzae*, and *Candida albicans* | Voriconazole injection for 3 days followed by caspofungin acetate injection for 6 days |
| 3 | Severe | *Pneumocystis jirovecii* | None |
| 4 | Severe | *Aspergillus fumigatus* | Voriconazole injection for 12 days, followed by posaconazole injection for 12 days and inhalation of AmBisome for 14 days |
| 5 | Non-severe | *Candida albicans* | None |

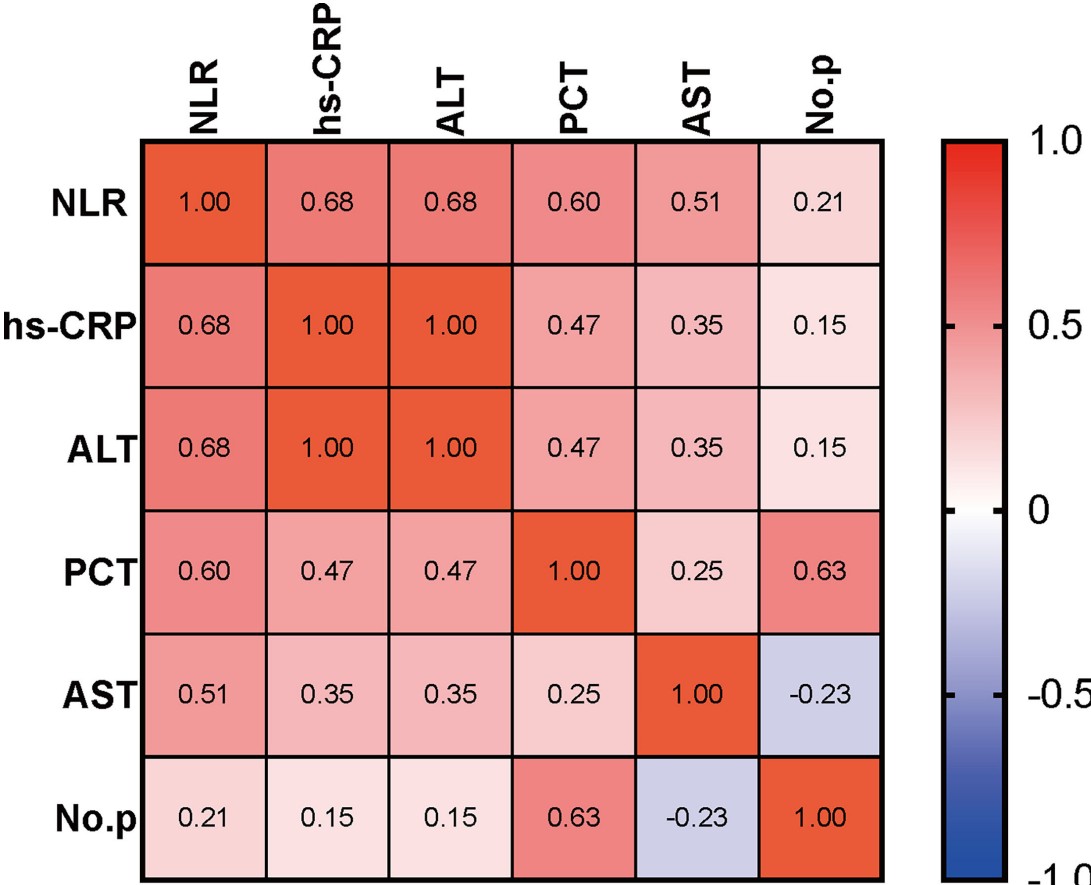

**FIG 3** The heatmap of the correlation matrix visualizes the correlation analysis of NLR, PCT, hs-CRP, ALT, AST, and the number of pathogens (No.p) detected by mNGS. The redder patches symbolize a higher correlation, and the bluer patches symbolize a lower correlation.

Psittacosis, caused by infection with *Chlamydia psittaci*, is an important zoonotic disease. The detection rate of *Chlamydia psittaci* by traditional methods, such as isolation and culture, may be difficult and time-consuming due to the intracellular bacterium. With the rapid development of mNGS and tNGS, the detection rate of *Chlamydia psittaci* has increased, and a growing number of cases have been reported in the USA, China, Germany, etc (18–20).

Our study found that 11 (45.8%) cases had a definite exposure to birds or poultry, which was consistent with previous studies (21, 22). However, some results in our study differed from those of previous studies. For example, patients with environmental exposure to wild ducks were seen in 16.7% of patients and 25.0% of patients with severe conditions in our study, while previous studies often combined duck and chicken

**TABLE 6** Spearman's correlation analysis between *Chlamydia psittaci* sequencing metrics (reads and abundance) and clinical indicators[a]

| Clinical variable | Spearman's Rho (ρ) (Reads) | P-value (Reads) | Spearman's Rho (ρ) (Abundance) | P-value (Abundance) |
|---|---|---|---|---|
| NLR | 0.459 | 0.0316 | 0.440 | 0.0404 |
| hs-CRP | 0.452 | 0.0346 | 0.368 | 0.0917 |
| PCT | 0.370 | 0.0985 | 0.483 | 0.0265 |
| Severe condition | 0.361 | 0.098 | 0.346 | 0.114 |
| Transfer to secondary hospitals post-discharge | 0.197 | 0.381 | 0.453 | 0.034 |

[a]NLR, neutrophil-to-lymphocyte ratio; hsCRP, hypersensitive C-reactive protein; PCT, procalcitonin.

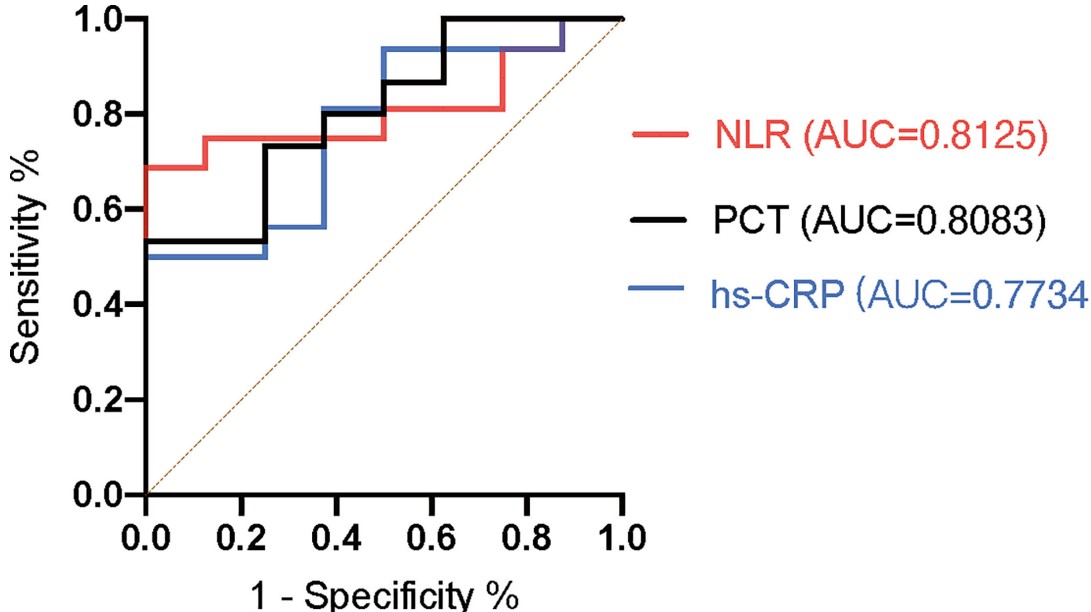

**FIG 4** The receiver operating characteristic (ROC) curves for predictive markers of identifying patients with severe psittacosis. The ROC curve for NLR is presented in the red line; the ROC curve for PCT is presented in the black line; the ROC curve for hs-CRP is presented in the blue line. AUC: area under the curve.

as poultry instead of listing separate ducks. Besides, the proportion of exposure to parrots in our study (16.7%) was higher when compared to the previous study (5.4%) (21), implying that the culturally or geographically factors could possibly influence the exposure of birds or poultry (12, 23). Similar to previous studies, our study found that the median time from onset of illness to diagnosis was about 10 days, and the long duration may explain why patients could not recall a clear history of exposure (22, 24).

Due to the similar and nonspecific clinical symptoms to other pneumonias, psittacosis is often underestimated or misdiagnosed in the beginning (25). As reported, *Chlamydia psittaci* could invade widespread epithelial and mononuclear-macrophage cells, causing a series of clinical manifestations. Consistent with previous studies, our study reveals that common symptoms of psittacosis mainly include fever, cough, fatigue, dyspnea, headache, and vomiting, similar to diseases relevant to the nervous and digestive systems (26). Due to headache, dizziness, or disorder of consciousness, several patients were initially admitted to the department of neurology and underwent lumbar puncture. No significant difference was observed between the patients of the severe group and those of the non-severe group, implying the difficulty in distinguishing between the severe group and the non-severe group only from clinical symptoms.

Laboratory tests showed increased WBCs, neutrophils, NLR, hs-CRP, and PCT in patients with severe psittacosis than that of the non-severe group and decreased Hb and PaO2/FiO2. Approximately 92% of the patients presented with hyperfibrinogenemia, 83% with liver damage, and 71% with hypoalbuminemia, which were different from previous data (21, 27). The levels of liver enzymes such as ALT and AST were not significantly different in the two groups, while the levels were higher than those in several studies. Notably, the level of ALT in our study was about 110 U/L, a nearly twofold change compared with the reported range from 46 to 63.5 U/L (24, 28), revealing that

**TABLE 7** Multivariate regression analysis of factors associated with the severe group[a]

| Variables | OR | 95% CI | P value |
| --- | --- | --- | --- |
| NLR | 1.103 | 1.001–1.216 | 0.048 |
| Age | | | 0.342 |
| Albumin | | | 0.974 |

[a]NLR, neutrophil-to-lymphocyte ratio; OR, odds ratio; CI, confidence interval.

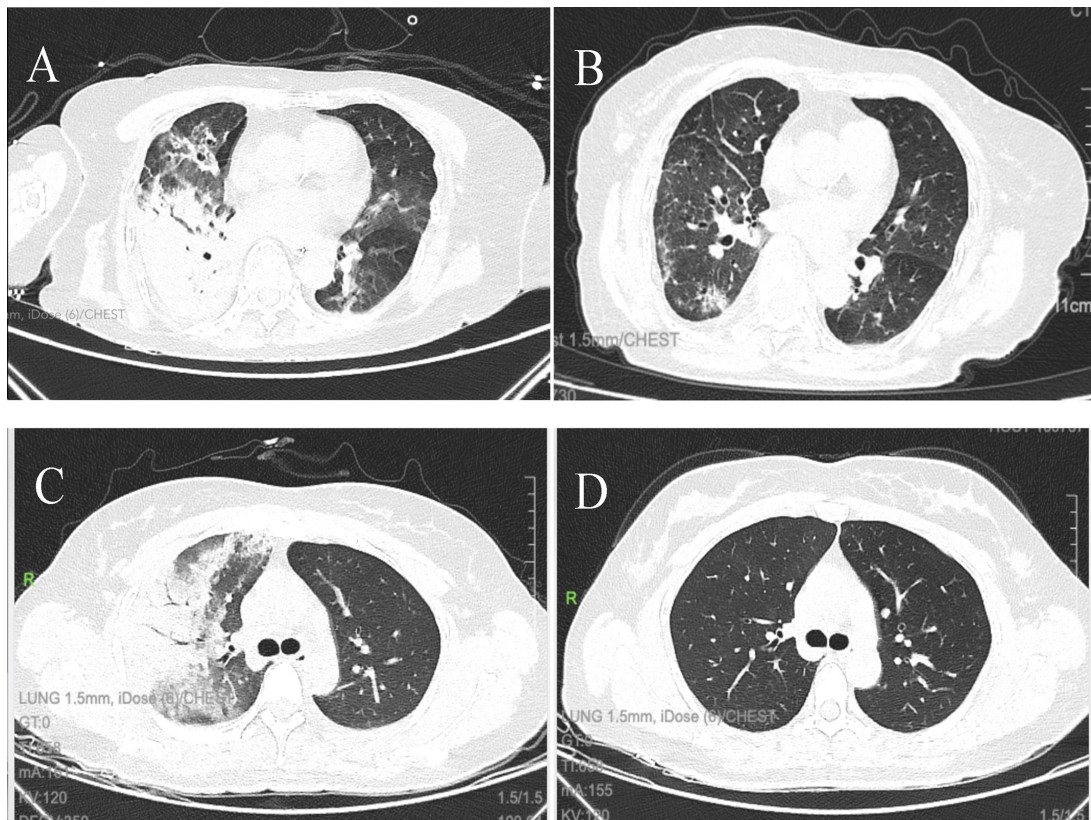

**FIG 5** Chest computed tomography (CT) findings of *Chlamydia psittaci* pneumonia. Serial chest CT scans of a 72-year-old woman with severe psittacosis (A) One day after hospital admission. (B) Six months after discharge, a repeated CT scan shows almost complete resolution of lesions but residual cord shadow. Serial chest CT scans of a 36-year-old woman with non-severe psittacosis. (C) One day after hospital admission. (D) One month after discharge, a repeated CT scan shows almost complete resolution without residual cord shadow.

patients in our study are more likely to be involved in liver injury, and a protective strategy for maintaining liver function is needed in clinical practice. The inflammatory markers, such as NLR, PCT, and hs-CRP, in our severe group were higher, indicating mixed or complex infection with multiple pathogens.

The mNGS results from different samples confirmed the mixed infection hypothesis. In our study, 62.50% of patients had mixed infection, and the coinfections were primarily related to viruses, different from another study from China, which reported the co-infection rate was 41.30% (29). Notably, the mNGS detected multiple kinds of pathogens and exhibited the reads of all pathogens, which warrants physicians to distinguish the real pathogens, opportunistic pathogens, resident flora, and contaminating flora and make appropriate decisions based on their professional knowledge. The rate of coinfections was 87.50% in patients of the severe group, and the main pathogens included herpes simplex virus (HSV), Epstein–Barr virus (EBV), intestinal flora, and *Candida albicans*, which partially agreed with previous research (18, 29). There was no significant difference in the coinfection rate between the severe and non-severe groups; however, the incidence of coinfection with fungi was higher in patients of the severe group than that of the non-severe group, which may lead to more complex conditions and adverse outcomes. Many studies have reported adverse outcomes such as death in severe groups (12, 26, 30), while none of our research participants died during hospitalization, which may be related to the multiple treatments of psittacosis in our research.

Oxygen support plays an important role in maintaining the respiratory and circulatory system (31, 32). In our research, invasive ventilation was commonly applied in severe groups, and a nasal cannula was commonly employed in non-severe groups.

The frequency and methods of respiratory support vary among studies. In four studies containing 53 patients, 39 patients, 27 patients, and 20 patients with severe psittacosis, the frequency of invasive ventilation was 39.6%, 41.0%, 44.4%, and 20.0% (21, 28, 29, 33). The rate of invasive ventilation in our research was about 88%, which was higher than the reported data and may lead to a longer duration of hospitalization. Additionally, ECMO is an advanced form of life support used primarily for patients with severe respiratory or cardiac failure who have failed conventional treatment, and two (2/8, 25%) patients of the severe group received V-V ECMO therapy in our study. Previous studies mentioned several cases of ECMO application in psittacosis, revealing the severity of psittacosis (26, 34).

Owing to the nonspecific and multiple symptoms, empirical antibiotic treatment, including cephalosporins and quinolones, was commonly prescribed before confirmation of psittacosis, which is not sufficiently effective. After diagnosis based on mNGS, tetracyclines such as doxycycline and omadacycline are recommended as first-line antibiotics. Other effective drugs include macrolides and quinolones. Due to the high resistance rate of common pathogens to macrolides, the use of azithromycin was lower than that of quinolones and tetracyclines. In a study of 27 patients with psittacosis, most (65.4%) patients received quinolones due to a lack of tetracyclines in the hospital (24). Several studies reported that the major antibiotics were tetracyclines with or without combination with quinolones (29). In a study of 122 subjects with psittacosis, 56.6% received tetracyclines and 16.4% received tetracyclines and quinolones, and the severe group showed a higher rate of tetracyclines and quinolones but a lower rate of quinolones compared to the non-severe group (22). Our research exhibited high proportions of tetracyclines, including minocycline (58.3%), omadacycline (50%), and doxycycline (8.3%), which partially agreed with the results of previous studies. In a study of 74 patients with psittacosis, doxycycline was the primary antibiotic, with a rate of 73%, higher than our data (21). The difference in doxycycline use may be related to the enrollment year. The previous study enrolled patients before March 2022; however, the use of omadacycline was not approved in China for treating CAP (December 2021) at that time, and doxycycline was more frequently applied in China.

In terms of comparison between groups, no significant difference was observed in the proportion of antibiotics between the two groups. However, the duration of omadacycline was longer in the severe group than in the non-severe group, which was rarely reported before. It is difficult to conclude the efficacy of a particular drug since mNGS could provide clues for diagnosis, but fails to analyze the sensitivity and resistance to the psittacosis species.

This study has several limitations. First, the relatively small sample size limits the statistical power, especially for detecting moderate associations or differences across subgroups, and increases the risk of type II errors. Second, the retrospective design introduces inherent selection bias. Our study was conducted in a tertiary care teaching hospital that predominantly admits complex or severely ill patients who cannot be managed at local or community hospitals. As a result, our cohort may not be representative of the broader population with community-acquired pneumonia (CAP), potentially limiting the generalizability of our findings. Third, while we focused on mNGS-based pathogen detection, we acknowledge that other confirmatory approaches, such as conventional PCR and serological testing, should be incorporated to validate these results. Last but not least, given that access to rapid mNGS testing and newer antimicrobials may differ across healthcare settings, our findings—especially those concerning treatment efficacy in severe versus non-severe cases—should be interpreted with caution. Further prospective, multicenter studies are needed to confirm these findings in more diverse and representative populations.

## Conclusion

In summary, a history of bird or duck contact and multiple symptoms with high fever, cough, and nervous and digestive symptoms could be suggestive of psittacosis. Patients

with psittacosis have a high rate of severe disease, and NLR is superior in efficacy to PCT and hs-CRP in detecting severe conditions. In our study, oxygen support and tetracyclines led to a good prognosis. Our findings provide some useful information for citizens to reduce exposure to ducks and parrots and help clinicians raise their awareness of severe psittacosis from clinical features and carry out effective treatments.

## ACKNOWLEDGMENTS

This research was supported by the National Natural Science Foundation of China (81970031, 81770031, and 82400029), the Key Research and Development Project of Jiangsu Province (BE2020616), and the China Postdoctoral Science Foundation (2024M761210).

T.X., Q.Y., and J.W. were responsible for methodology, investigation, review, extraction of data and editing, original draft writing, and formal analysis. Z. Wu, Z.C., Z. Wang, W.S., and M.Z. were responsible for review and editing, methodology, and resources. N.J. and M.H. were responsible for conceptualization, supervision, project administration, and funding acquisition.

## AUTHOR AFFILIATION

[1]Department of Respiratory and Critical Care Medicine, The First Affiliated Hospital of Nanjing Medical University, Nanjing, China

## AUTHOR ORCIDs

Tingting Xu http://orcid.org/0000-0001-9892-7952
Wenkui Sun http://orcid.org/0000-0002-2992-9783
Mingshun Zhang http://orcid.org/0000-0001-5925-0168
Ningfei Ji http://orcid.org/0000-0002-5044-2240

## FUNDING

| Funder | Grant(s) | Author(s) |
| --- | --- | --- |
| National Natural Science Foundation of China | 81970031, 81770031, 82400029 | Tingting Xu |
| | | Mao Huang |
| Jiangsu Provincial Key Research and Development Program | BE2020616 | Mao Huang |
| China Postdoctoral Science Foundation | 2024M761210 | Tingting Xu |

## AUTHOR CONTRIBUTIONS

Tingting Xu, Conceptualization, Data curation, Investigation, Methodology, Validation, Writing – original draft | Qi Yuan, Conceptualization, Data curation, Formal analysis, Methodology, Visualization, Writing – original draft | Jiayue Wang, Data curation, Investigation, Validation, Writing – review and editing | Zhenzhen Wu, Data curation, Investigation, Validation, Writing – review and editing | Zhongqi Chen, Formal analysis, Investigation, Software, Validation | Zhengxia Wang, Formal analysis, Methodology, Validation, Writing – review and editing | Wenkui Sun, Investigation, Methodology, Validation, Writing – review and editing | Mingshun Zhang, Investigation, Methodology, Validation, Writing – review and editing | Ningfei Ji, Funding acquisition, Supervision, Validation, Writing – review and editing | Mao Huang, Conceptualization, Funding acquisition, Supervision

## ETHICS APPROVAL

This study was conducted following the Declaration of Helsinki and was approved by the Ethics Committee of the First Affiliated Hospital of Nanjing Medical University

with ethical approval reference number IRB-2024-SR-678. Written informed consent was waived because this study was retrospective.

## ADDITIONAL FILES

The following material is available online.

## Open Peer Review

**PEER REVIEW HISTORY (review-history.pdf).** An accounting of the reviewer comments and feedback.

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
