## [Reviewer comments · Microbiology Spectrum]

Microbiology Spectrum

Clinical characteristics and risk factors associated with severe community-acquired pneumonia infected by *Chlamydia psittaci*

Tingting Xu, Qi Yuan, Jlayue Wang, Zhenzhen Wu, Zhongqi Chen, Zhengxia Wang, wenkui sun, Mingshun Zhang, Ningfei Ji, and Mao Huang

Corresponding Author(s): Mao Huang, The First Affiliated Hospital of Nanjing Medical University

Review Timeline:

Submission Date:	February 15, 2025
Editorial Decision:	May 4, 2025
Revision Received:	July 18, 2025
Accepted:	September 1, 2025

Editor: Max Maurin

Reviewer(s): The reviewers have opted to remain anonymous.

Transaction Report:

DOI: <https://doi.org/10.1128/spectrum.00477-25>

Re: Spectrum00477-25 (Clinical characteristics and risk factors associated with severe community-acquired pneumonia infected by *Chlamydia psittaci*)

Dear Dr. Mao Huang:

Thank you for the privilege of reviewing your work. Below you will find my comments, instructions from the Spectrum editorial office, and the reviewer comments.

Both reviewers suggested major modifications. They mainly concern the need to clarify certain methodological data, to strengthen the results section and their statistical analysis, which should lead to modifications at the level of the discussion. It is also necessary to improve the presentation of references and edit the English of the manuscript.

Revision Guidelines

Sincerely,
Max Maurin
Editor
Microbiology Spectrum

Reviewer #1 (Comments for the Author):

Xu and colleagues present a descriptive retrospective review of 24 patients with psittacosis to identify risk factors.

Specific comments

1. Lines 44 - 47 and references. The pathologic lesion depends on the immune response to the organism and not its differentiation to the infectious elementary body. The reference numbers in the reference section were impossible to read because of the line numbers.
2. Lines 49 - 50. Lysosomes do not phagocytize.
3. Lines 56 - 60. Why would metagenomic sequencing be more rapid and accurate than a highly specific PCR?
4. Lines 189 - 190. What are the performance characteristics of the mNGS? Perhaps these are detailed in a reference but, as previously mentioned, the line number obscured the reference numbers.
5. Lines 382 - 289. Could these differences be attributed to differences in culture? And, if so, could the risk factors change by geography and culture.

Reviewer #2 (Comments for the Author):

Major Comments

1. Sample Size and Statistical Power

The study includes only 24 patients (8 severe, 16 non-severe), which may result in insufficient statistical power.

Recommendation: Incorporate baseline matching—such as propensity score matching on key covariates (e.g., age, comorbidities)—to mitigate confounding. Additionally, apply cross-validation to evaluate the robustness of your predictive model.

2. Pathogen Load Analysis

Although pathogen read counts and relative abundances from mNGS are reported, the manuscript lacks an analysis of how *Chlamydia psittaci* burden correlates with clinical outcomes or inflammatory biomarkers.

Recommendation: Perform correlation analyses between pathogen load and both clinical endpoints and inflammatory markers to elucidate the impact of microbial burden on disease phenotype.

Minor Comments

Methods

1. Sample Selection Criteria

The manuscript states that mNGS was performed on BALF, blood, and sputum samples but does not describe how each sample type was chosen.

Recommendation: Specify the inclusion criteria or decision algorithm for selecting each specimen type.

2. Handling of Missing Data and Outliers

With a small cohort, individual missing values or extreme outliers may disproportionately influence results.

Recommendation: Define how missing values were imputed and outline your criteria for identifying and managing outliers.

Discussion

The current "Limitations" paragraph should more explicitly address issues arising from the small sample size, such as reduced statistical power, potential selection bias (e.g., enrollment of complex cases at tertiary centers), and restricted generalizability of the findings.

Recommendation: Expand this section to acknowledge these concerns and discuss their potential effects on study conclusions.

Responses to Review Comments on Manuscript (Spectrum00477-25)
"Clinical characteristics and risk factors associated with severe community-acquired pneumonia infected by *Chlamydia psittaci* "

Dear Editor:

We sincerely thank you and the reviewers for your time and effort in reviewing this paper. We have carefully addressed all the comments, and the corresponding amendments are updated in the revised manuscript, which is marked in red. The detailed point-by-point responses are summarized as follows:

Response to Comments from Reviewer 1

Xu and colleagues present a descriptive retrospective review of 24 patients with psittacosis to identify risk factors.

Thanks very much for your valuable comments and suggestions on this paper. We have carefully addressed all your comments, and the detailed responses for each comment are summarized as follows:

Comment 1-1: *Lines 44 - 47 and references. The pathologic lesion depends on the immune response to the organism and not its differentiation to the infectious elementary body. The reference numbers in the reference section were impossible to read because of the line numbers.*

Response: Thank you very much for your invaluable feedback. First, we appreciate your clarification regarding the pathogenesis. You are absolutely right. The pathologic lesion depends on the exaggerated immune response to the elementary body (EB), not the differentiation process itself. We have revised the relevant sentence to accurately reflect this mechanism and to eliminate the previous ambiguity in the revised manuscript.

Second, we sincerely apologize for the difficulty in reading the reference numbers due to overlapping line numbers. This was a formatting issue in the original submission PDF. We have corrected the reference formatting in the revised version to ensure clarity and readability.

We thank you again for helping us improve the precision and presentation of our manuscript.

Comment 1-2: *Lines 49 - 50. Lysosomes do not phagocytize.*

Response: Thank you for your insightful comments.

We agree that lysosomes do not themselves perform phagocytosis. To clarify, mononuclear phagocytes (i.e., monocytes and macrophages) are the primary cellular hosts for *C. psittaci*, serving both as replication niches and immunologic sentinels. In contrast, lysosomes are intracellular organelles that play a critical role in the degradation of phagocytosed material, and are frequently subverted by *C. psittaci* to evade immune clearance.

To correct the mechanistic misstatement, we have revised the sentence to explicitly refer to "mononuclear phagocytes" as the cells responsible for phagocytosis and to "lysosomal degradation" as the immune process involved. This revision better reflects the cellular biology and avoids mischaracterizing lysosomal function.

Comment 1-3: *Lines 56 - 60. Why would metagenomic sequencing be more rapid and accurate than a highly specific PCR?*

Response: We appreciate your insightful query regarding the comparative advantages of metagenomic next-generation sequencing (mNGS) over highly specific PCR. In brief, mNGS and a highly specific PCR have separate advantages and applications in clinical practice. First, traditional PCR requires a priori knowledge of the pathogen to design primers (*ompA* for *C. psittaci*). Second, PCR detects the pathogens with more than 10^3 copies, leading to a low sensitivity in the earlier phase of infection. Third, as reported in our study and previous research, suspected cases with atypical presentations often need sequential PCR assays for multiple pathogens (e.g., viruses and fungi), delaying diagnosis of multiple infections. However, PCR remains the gold standard for targeted diagnosis where pathogen identity is clinically suspected due to the high specificity. On the other hand, mNGS could generate reports about 24 hours after sample detection on the basis of untargeted sequencing of numerous nucleic acids of common pathogens and automated workflows to identify millions reads from pathogens.

Overall, PCR is well-suited for screening during outbreaks or epidemics of known pathogens, as exemplified by its use with throat swab samples during the COVID-19 pandemic. Conversely, mNGS is primarily indicated for critically ill patients, immunocompromised

individuals, or cases of fever of unknown origin. When mNGS yields positive results, PCR can subsequently be employed for confirmatory verification. To prevent potential misinterpretation, we have removed the original ambiguous sentence, and we have revised the manuscript to reflect this more nuanced comparison between mNGS and PCR.

Comment 1-4: *Lines 189 - 190. What are the performance characteristics of the mNGS? Perhaps these are detailed in a reference but, as previously mentioned, the line number obscured the reference numbers.*

Response: We appreciate your insightful commentary. Metagenomic next-generation sequencing (mNGS) provides an untargeted, hypothesis-free approach for pathogen detection, enabling comprehensive microbial profiling and identification of a diverse array of pathogens directly from clinical specimens. This capability renders mNGS particularly valuable for detecting rare, novel, or unexpected pathogens, as well as culture-negative infections following antimicrobial therapy. However, the results of mNGS require careful interpretation because of the high sensitivity of detection of commensals and environmental contaminants in clinical practice. Overall, the mNGS holds significant promise as a routine component of precision infectious disease management; its optimal utility requires integration with complementary diagnostic methodologies. We have incorporated relevant references into the section detailing "The workflow of mNGS (sample processing, DNA/RNA extraction, construction of DNA/RNA libraries, sequencing and bioinformatic analysis)" ensured their unambiguous citation. In terms of reference numbers, we sincerely apologize for the difficulty in reading the reference numbers due to overlapping line numbers. This was a formatting issue in the original submission PDF. We have corrected the reference formatting in the revised version to ensure clarity and readability as you guided before.

We thank you again for pointing out this important clarification, which we believe improves the clarity and completeness of the manuscript.

Comment 1-5: *Lines 382 - 289. Could these differences be attributed to differences in culture? And, if so, could the risk factors change by geography and culture.*

Response: Thank you for your invaluable comments. We agree that the observed discrepancies in exposure history may indeed reflect

cultural-geographical influences, evidenced through three key aspects. First, free-range duck husbandry remains common in rural China (vs. industrialized farming in the West), elevating wild duck exposure. Second, urban pet parrot ownership in East Asia increases parrot-associated risks. Third, cultural perceptions of "bird exposure" cause recall bias (e.g., underreported live poultry market contacts in China), while earlier mNGS adoption in developed regions improves detection in cases without recalled exposure.

Geography-dependent risk profiles are evidenced by higher infection risk in Dutch pigeon breeders and Australian pet cockatoos as primary sources, validating culturally/geographically variable risk factors; we have incorporated these perspectives into the section of Discussion to emphasize that effective psittacosis control requires context-specific public health messaging, risk assessment, and diagnostic awareness.in the revised manuscript.

Once again, we sincerely thank you for your time and valuable input, which have helped us improve this paper. Should you have any further suggestions or comments, please do not hesitate to share them with us. We are eager to enhance the manuscript's content to ensure its excellence and contribution to the field.

Response to Comments from Reviewer 2

Thank you for your invaluable insights and recommendations regarding this paper. We have carefully taken into account each of your comments and suggestions. To succinctly encapsulate our responses to each comment, we offer the following summarization:

Comment 2-1: *Sample Size and Statistical Power. The study includes only 24 patients (8 severe, 16 non-severe), which may result in insufficient statistical power. Recommendation: Incorporate baseline matching-such as propensity score matching on key covariates (e.g., age, comorbidities)-to mitigate confounding. Additionally, apply cross-validation to evaluate the robustness of your predictive model.*

Response: We sincerely appreciate the reviewer's valuable feedback regarding sample size and statistical power concerns. In response to the recommendation for propensity score matching (PSM), we performed analysis using key covariates including age and gender; however, the limited cohort size (N=24) yielded only 4 matched pairs after standard caliper matching, which paradoxically exacerbated confounding imbalances and introduced selection bias-consistent with methodological literature indicating PSM requires ≥ 50 samples for reliable implementation. Given these constraints and our focus on developing early prediction tools for a rare, severe disease (with consecutively enrolled, clinically confirmed cases), the sample size aligns with similar studies in this challenging research area [1, 2].

In terms of cross-validation, given that the current dataset contains only 24 samples, performing conventional k-fold cross-validation (e.g., 5-fold or 10-fold) would lead to extremely small test subsets (e.g., 2–5 samples per fold), which could result in unstable and high-variance estimates of model performance. We further note no significant baseline differences in age ($p=0.079$) or gender ($p=0.673$) between non-severe and severe groups. While we fully acknowledge the limitations inherent to small-sample observational studies, we believe these methodological considerations and additional analyses address the concerns regarding confounding mitigation and model robustness.

Reference:

1. Chen X, Cao K, Wei Y, et al. Metagenomic next-generation sequencing in the diagnosis of severe pneumonias caused by *Chlamydia psittaci*. *Infection*. 2020;48(4):535-42.

2. Su S, Su X, Zhou L, et al. Severe *Chlamydia psittaci* pneumonia: clinical characteristics and risk factors. *Ann Palliat Med*. 2021;10(7):8051-60.

Comment 2-2: *Pathogen Load Analysis. Although pathogen read counts and relative abundances from mNGS are reported, the manuscript lacks an analysis of how Chlamydia psittaci burden correlates with clinical outcomes or inflammatory biomarkers. Recommendation: Perform correlation analyses between pathogen load and both clinical endpoints and inflammatory markers to elucidate the impact of microbial burden on disease phenotype.*

Response: Thank you for your invaluable comment. Following your guidance, we collected sequencing metrics (primarily reads and relative abundance), and we included 22 patients. Two patients were excluded: one with only blood samples and another diagnosed in the nephrology department, both of whom lacked specific clinical records necessary for analysis.

Since the reads and relative abundance of *Chlamydia psittaci* exhibited non-normal distributions, and clinical outcomes (severe pneumonia status and post-discharge transfer to secondary hospitals) were dichotomized, we employed Spearman rank correlation to evaluate associations between *Chlamydia psittaci* sequencing metrics (reads, relative abundance) and inflammatory markers (NLR, PCT, hs-CRP) or clinical outcomes. Correlation coefficients are reported as Spearman's Rho (ρ), with detailed results presented in Table 6. The correlation analysis revealed that both reads and relative abundance of *Chlamydia psittaci* were positively correlated with NLR (reads: $\rho=0.46$, $P=0.032$; relative abundance: $\rho=0.44$, $P=0.040$). Notably, the relative abundance also showed a positive correlation with the clinical endpoint of post-discharge transfer to secondary hospitals ($\rho=0.45$, $P=0.034$). However, we acknowledge that sequencing reads and relative abundance do not directly equate to actual pathogen burden; the absence of qPCR-validated absolute quantification (in copies/mL) limits our ability to draw definitive conclusions about the relationship between true pathogen load and clinical variables. In future studies, we will pay more attention to the absolute quantification of pathogen load to address this limitation.

We have updated the manuscript to include correlation analyses between reads, abundance, and clinical variables as described. We sincerely appreciate your insightful suggestions, which have significantly strengthened the rigor of our analysis.

Table 6 Spearman correlation analysis between *chlamydia psittaci* sequencing metrics (reads and abundance) and clinical indicators

Clinical Variable	Spearman's Rho (ρ)	p-value	Spearman's Rho (ρ)	p-value
	(Reads)	(Reads)	(Abundance)	(Abundance)
NLR	0.459	0.0316	0.440	0.0404
hs-CRP	0.452	0.0346	0.368	0.0917
PCT	0.370	0.0985	0.483	0.0265
Severe condition	0.361	0.098	0.346	0.114
Transfer to secondary hospitals	0.197	0.381	0.453	0.034

Comment 2-3: *Sample Selection Criteria. The manuscript states that mNGS was performed on BALF, blood, and sputum samples but does not describe how each sample type was chosen. Recommendation: Specify the inclusion criteria or decision algorithm for selecting each specimen type.*

Response: We greatly appreciate your detailed feedback. Regarding sample selection, bronchoalveolar lavage fluid (BALF) was designated as the primary specimen type for metagenomic next-generation sequencing (mNGS) in this study, based on its high diagnostic yield in lower respiratory tract infections. However, due to clinical or patient-specific limitations, a few deviations occurred: Two patients declined BALF collection; in these cases, peripheral blood was used as an alternative specimen. One patient provided all three sample types (BALF, blood, and sputum), allowing comparative analysis. Another patient contributed both BALF and blood specimens for evaluation. The remaining 20 patients underwent mNGS exclusively on BALF samples, as summarized in Table 4.

We have now clarified these inclusion criteria and the rationale for specimen selection in the revised Methods section (2.3 Criteria for diagnosis and grouping), to ensure transparency in sample-level decision-making and clinical applicability.

Comment 2-4: *Handling of Missing Data and Outliers. With a small cohort, individual missing values or extreme outliers may disproportionately influence results. Recommendation: Define how missing values were imputed and outline your criteria for identifying and managing outliers.*

Response: Thank you for emphasizing the critical importance of rigorous missing data and outlier management in our small cohort study. In

our small cohort study, missing values exhibited non-random patterns aligned with clinical protocols (e.g., stable patients omitting blood gas analysis; non-diabetics foregoing HbA1c testing). Consequently, these variables were retained in univariate analyses but excluded through listwise deletion in multivariate regression. For outliers ($>3\times$ IQR beyond quartiles or exceeding clinical thresholds), all instances underwent blinded verification by two clinicians. Biologically plausible outliers were retained and analyzed via Huber regression to minimize leverage. Comprehensive robustness analyses confirmed consistent conclusions across all methodological approaches. We have supplemented the detailed methodology in the "Statistical Analysis" section of the revised manuscript. Thank you for your suggestion, which has further enhanced the rigor of the study.

Comment 2-5: *Discussion. The current "Limitations" paragraph should more explicitly address issues arising from the small sample size, such as reduced statistical power, potential selection bias (e.g., enrollment of complex cases at tertiary centers), and restricted generalizability of the findings. Recommendation: Expand this section to acknowledge these concerns and discuss their potential effects on study conclusions.*

Response: Thank you very much for your invaluable suggestion. We fully agree that a more explicit and structured discussion of the implications of our small sample size and potential selection bias is necessary to accurately contextualize the findings. In response, we have substantially revised the "Limitations" paragraph in the Discussion section as follows:

This study has several limitations. First, the relatively small sample size limits the statistical power, especially for detecting moderate associations or differences across subgroups, and increases the risk of type II errors. Second, the retrospective design introduces inherent selection bias. Our study was conducted in a tertiary teaching hospital that predominantly admits complex or severely ill patients who cannot be managed at local or community hospitals. As a result, our cohort may not be representative of the broader population with community-acquired pneumonia (CAP), potentially limiting the generalizability of our findings. Third, while we focused on mNGS-based pathogen detection, we acknowledge that other confirmatory approaches, such as conventional PCR and serological testing, should be incorporated to validate these results. Finally, because access to rapid mNGS testing and newer antimicrobials may

differ across healthcare settings, our findings—especially those concerning treatment efficacy in severe versus non-severe cases—should be interpreted with caution. Further prospective, multicenter studies are needed to confirm these findings in more diverse and representative populations.

Once again, we sincerely thank you for your time and valuable input, which have helped us improve this paper. Should you have any further suggestions or comments, please do not hesitate to share them with us. We are eager to enhance the manuscript's content to ensure its excellence and contribution to the field.

Re: Spectrum00477-25R1 (Clinical characteristics and risk factors associated with severe community-acquired pneumonia infected by *Chlamydia psittaci*)

Dear Dr. Mao Huang:

Thank you for responding thoroughly and in detail to all of the comments from both reviewers. The manuscript has been significantly improved.

Your manuscript has been accepted, and I am forwarding it to the ASM production staff for publication. Your paper will first be checked to make sure all elements meet the technical requirements. ASM staff will contact you if anything needs to be revised before copyediting and production can begin. Otherwise, you will be notified when your proofs are ready to be viewed.

Sincerely,
Max Maurin
Editor
Microbiology Spectrum

Reviewer #2 (Comments for the Author):

The author has already made the necessary revisions and there are no more issues.